# Stromal Senescence following Treatment with the CDK4/6 Inhibitor Palbociclib Alters the Lung Metastatic Niche and Increases Metastasis of Drug-Resistant Mammary Cancer Cells

**DOI:** 10.3390/cancers15061908

**Published:** 2023-03-22

**Authors:** Gregory T. Gallanis, Ghada M. Sharif, Marcel O. Schmidt, Benjamin N. Friedland, Rohith Battina, Raneen Rahhal, John E. Davis, Irfan S. Khan, Anton Wellstein, Anna T. Riegel

**Affiliations:** Lombardi Comprehensive Cancer Center, Georgetown University Medical Center, Washington, DC 20007, USA

**Keywords:** CDK4/6 inhibitors, palbociclib, abemaciclib, lung metastasis, mammary cancer, senescence, endothelium, p16-INK-ATTAC, senolytic, macrophage, monocyte, p16^INK4A^, p21^CIP1^, TGF-β

## Abstract

**Simple Summary:**

CDK4/6 inhibitors (CDKis) are a first-line treatment for metastatic hormone-receptor-positive breast cancer, but resistance frequently develops. One potential resistance mechanism could be induction of cellular senescence in non-tumor tissues. The aim of our study was to identify CDKi-induced changes to host tissues that impact metastasis. Using mouse models, we found that pretreatment with palbociclib can increase metastatic seeding of CDKi-resistant mammary cancer cells in lungs and that this can be mitigated by eliminating senescent host cells. We describe palbociclib-induced gene expression changes in lungs that correlate with this effect and reveal altered intra-lung immune populations. Senescent endothelial cells are identifiable within lung metastases of mice pretreated with palbociclib. Palbociclib-treated primary endothelial cell lines become senescent and increase tumor cell migration and monocyte trans-endothelial invasion. These studies describe how CDKi-induced cellular senescence in host tissues could affect metastasis in breast cancer, which remains a key obstacle to achieving long-term survival.

**Abstract:**

Background: CDK4/6 inhibitors (CDKi) have improved disease control in hormone-receptor-positive, HER2-negative metastatic breast cancer, but most patients develop progressive disease. Methods: We asked whether host stromal senescence after CDK4/6 inhibition affects metastatic seeding and growth of CDKi-resistant mammary cancer cells by using the p16-INK-ATTAC mouse model of inducible senolysis. Results: Palbociclib pretreatment of naïve mice increased lung seeding of CDKi-resistant syngeneic mammary cancer cells, and this effect was reversed by depletion of host senescent cells. RNA sequencing analyses of lungs from non-tumor-bearing p16-INK-ATTAC mice identified that palbociclib downregulates immune-related gene sets and gene expression related to leukocyte migration. Concomitant senolysis reversed a portion of these effects, including pathway-level enrichment of TGF-β- and senescence-related signaling. CIBERSORTx analysis revealed that palbociclib alters intra-lung macrophage/monocyte populations. Notably, lung metastases from palbociclib-pretreated mice revealed senescent endothelial cells. Palbociclib-treated endothelial cells exhibit hallmark senescent features in vitro, upregulate genes involved with the senescence-associated secretory phenotype, leukocyte migration, and TGF-β-mediated paracrine senescence and induce tumor cell migration and monocyte trans-endothelial invasion in co-culture. Conclusions: These studies shed light on how stromal senescence induced by palbociclib affects lung metastasis, and they describe palbociclib-induced gene expression changes in the normal lung and endothelial cell models that correlate with changes in the tumor microenvironment in the lung metastatic niche.

## 1. Introduction

Despite advances in screening, diagnosis, and treatment, approximately 30% of women diagnosed with early-stage breast cancer develop advanced or metastatic disease [1]. Metastatic breast cancer (MBC) is currently considered treatable but not curable, and the goals of therapy are to prolong survival, palliate disease-related symptoms, and improve quality of life. CDK4/6 inhibitors (CDKis), such as palbociclib, used in combination with endocrine therapy (ET), such as an aromatase inhibitor, are the first-line treatment of hormone-receptor-positive, HER2-negative MBC [2]. CDKis offer improved disease control, as progression-free survival is increased by the addition of a CDKi compared to ET alone; however, while there is some evidence that CDKis may slightly increase overall survival for some subgroups of breast cancer patients, not all patients respond to therapy, and most patients who do respond initially eventually develop therapeutic resistance and progressive disease [3,4,5,6,7,8,9]. It is unclear to what extent the host responses contribute to relapse, as CDK4/6 inhibition induces several potentially clinically relevant phenotypes in stromal cells in parallel to its direct effect on cancer cells [10]. One less-studied consequence of CDKi use in cancer is the increase in cellular senescence in the normal host tissue environment that is induced by this class of drug [11].

Cellular senescence is a condition of stable growth arrest that is thought to have evolved to control oncogenic progression [12,13]. Senescence is induced by multiple intrinsic stimuli such as DNA damage and metabolic perturbations or by extrinsic stimuli such as genotoxic drugs, ionizing radiation, and tissue injury. Senescent cells are known to produce a plethora of secreted proteins called the senescence-associated secretory phenotype (SASP), which can inhibit the malignant progression of cells but may also lead to an inflamed microenvironment that fosters enhanced metastasis and therapeutic resistance [14,15]. Host stromal cells that enter senescence as a consequence of systemic anticancer therapy are known to promote tissue degeneration and development of metastases, and their detrimental effects are mediated by the production of the inflammatory cytokines and chemokines of the SASP [16]. Furthermore, elimination of therapy-induced senescent host cells in animal models has been shown to alleviate adverse effects of therapy such as fatigue, myelosuppression, bone loss, and cancer progression [15,17]. CDKis have been shown to replicate many of the hallmark characteristics of cellular senescence, such as potent cell cycle arrest, increased immunogenicity, and expression of senescence-associated β-galactosidase (SA-β-gal) [18,19,20]. However, it is unclear to what extent the senescent states triggered in host cells by CDKis recapitulate the detrimental effects that are known to emerge when cells are induced into senescence by other mechanisms.

Here, we asked whether cellular senescence after CDKi treatment affects metastatic seeding and metastatic growth of breast cancer cells. To address this, we used the p16-INK-ATTAC mouse model of inducible senolysis, in which p16^INK4A^-expressing senescent host cells were selectively induced to apoptosis in vivo by a drug-induced switch [21]. In this model, palbociclib pretreatment increased metastatic seeding of syngeneic breast cancer cell lines, and this effect was ameliorated by elimination of senescent host cells. RNA sequencing analyses documented short- and long-term palbociclib-induced or combination-palbociclib-plus-senolytic-induced gene expression changes on stromal tissues as well as changes in tissue invasion by macrophages and monocytes.

## 2. Materials and Methods

### 2.1. Animal Models

Studies in mice were reviewed and approved by the Georgetown University Animal Care and Use Committee (protocol #2016-1113). p16-INK-ATTAC (“ATTAC”) mice were obtained as a gift from the van Deursen laboratory [21]. C3(1)-SV40/T-antigen-REAR (“REAR”) mice were obtained as a gift from Jefferey E. Green (National Cancer Institute; Bethesda, MD, USA) [22]. Female C57BL/6 (“C57”) mice were purchased from Jackson Labs. Animals were housed in a pathogen-free environment. C57 pretreatment experiments: Female C57 mice aged 8–12 weeks received oral gavage of 150 mg/kg palbociclib (MedChemExpress, Princeton, NJ, USA) or sterile water diluted in a water-based vehicle composed of 0.5% methylcellulose and 0.2% Tween 80 daily for 10 days (10 total doses). After completion of the 10-day treatment and 96 h drug washout period, mice were injected with 10^6^ E0771 cells (see Cell Culture section below) diluted in sterile PBS intravenously in the tail vein, monitored for 24 days, and then collected. ATTAC pretreatment experiments: Female ATTAC mice aged 8–12 weeks received oral gavage of 150 mg/kg palbociclib or sterile water (as above) daily for 10 days (10 total doses) and intraperitoneal injection of 2 mg/kg AP20187 (MedChemExpress Princeton, NJ, USA) or DMSO vehicle every three days for 10 days (four total doses). After completion of the 10-day treatment and 96 h drug washout period, mice were injected intravenously with 10^6^ E0771 cells as above, monitored for 15 days, and then collected. REAR pretreatment experiments: Female REAR mice aged 8–12 weeks received oral gavage of 150 mg/kg palbociclib or sterile water (as above) daily for 10 days (10 total doses). After completion of the 10-day treatment and 96 h drug washout period, mice were injected with 10^6^ m6-mCherry cells (see Cell Culture section below) diluted in sterile PBS intravenously in the tail vein, then monitored for 60 days and collected. Primary tumor experiments in ATTAC: 10^6^ E0771 cells were implanted subcutaneously in the right and left flanks of female ATTAC mice. Mice were then randomized to receive treatment with palbociclib, AP alone, palbociclib + AP, or vehicle control. One day after injection, treatment with 150 mg/kg daily palbociclib or vehicle control was initiated for 10 days (10 total doses). One day after completion of the palbociclib treatment, treatment with 2 mg/kg AP20187 (AP) or vehicle control was initiated every three days for 10 days (4 doses). Tumors were resected 21 days after implantation. Mice were then maintained for 7 days and then treated with 5 additional doses of AP 2 mg/kg or vehicle every three days before collection. Primary tumor experiments in REAR: 10^6^ m6 cells in 50% Corning Matrigel/PBS mix were implanted percutaneously into the right and left inguinal mammary glands of female REAR mice aged 8–12 weeks. Fourteen days after injection, treatment with 150 mg/kg daily palbociclib or vehicle control was initiated for 10 days (10 total doses). REAR mice tumors were collected 24 h after the last palbociclib dose. Tumor area (L × W) calculated by caliper measurements over the skin.

### 2.2. Histology

Hematoxylin and eosin (H&E) staining was performed on paraffin-embedded 5 µm sections of lungs. Whole slide images were captured using an Aperio GT450 at the Georgetown University Histopathology and Tissue Shared Resource Core and a Hamamatsu NanoZoomer 2.0-HT System at the Washington University in St. Louis Alafi Neuroimaging Lab. Immunohistochemistry staining against SV40 Large T antigen (BD, Franklin Lakes, NJ, USA, #554150) was performed as described previously [23] on m6 lung metastases from REAR mice. Metastasis burden as a proportion of cross-sectional tissue area and metastatic foci counts was calculated using QuPath v0.3.2 [24] in a blinded manner. Immunofluorescence staining was performed as described previously [25] on paraffin-embedded 5 µm sections of lungs for CD-31/Pecam-1 (Cell Signaling Technologies, Danvers, MA, USA, #77699), GFP (Invitrogen, Waltham, MA, USA, #332600), and Hoechst dye (Fisher Scientific, Waltham, MA, USA). Slides were imaged on a Zeiss LSM800 microscope.

### 2.3. Cell Culture

E0771 cells were obtained from Dr. Louis Weiner (Georgetown University). m6 cells were obtained from Dr. Jefferey E. Green (National Cancer Institute, Bethesda, MD, USA), and m6 cells expressing H2B-mCherry (“m6-mCherry”) were derived by Dr. Gray Pearson (Georgetown University Medical Center). THP-1 and MDA-MB-231 cells were obtained from the Georgetown University Tissue Culture Shared Resource. E0771, m6, m6-mCherry, MDA-MB-231, and THP-1 cells were cultured in Dulbecco’s Modified Eagle Medium (DMEM) (Gibco/Invitrogen, Carlsbad, CA, USA) supplemented with 10% fetal bovine serum. All cell lines were fingerprinted prior to use and tested for mycoplasma contamination regularly. Human umbilical vein endothelial cells (HUVEC) were purchased from Lonza and cultured in EGM-2 media (Lonza, Gampel, Switzerland) according to the manufacturer’s instructions.

### 2.4. Drug Sensitivity Assays

Effects of CDK4/6 inhibition in vitro were assessed using various concentrations of palbociclib or abemaciclib (Taizhou Crene Biotechnology, Taizhou, China); NMR was performed at the Georgetown University Department of Chemistry to confirm compound structure prior to use. For viability or cell size assays, cells were seeded for 24 h prior to treatment, and drug effects were assessed after 72 h of treatment. Viability was assessed using CellTiter-Glo (Promega, Madison, WI, USA) or by crystal violet stain. Cell size was assessed using images captured on an Olympus IX-71 inverted epifluorescence microscope and quantified using QuPath v0.3.2 [24].

### 2.5. Cell Cycle and Senescence Assays

HUVEC were passaged once after cryorecovery, cultured for 24 h, treated with indicated doses of palbociclib or abemaciclib in water vehicle for 72 h, and then collected for downstream processing. A separate aliquot of untreated HUVEC in exponential growth phase was collected alongside for reference comparisons. Cell cycle state was assessed for three independent replicates of one million cells per treatment condition, quantified using the Vendelov method [26]. Staining for senescence-associated β-galactosidase (SA-β-gal) was carried out on three independent replicates per treatment condition using the Cell Signaling Technologies SA-β-gal staining kit according to the manufacturer’s instructions. After developing the stain for 48 h, the cells were fixed using 4% paraformaldehyde in PBS and counterstained with Nuclear Fast Red (Thermo Scientific, Waltham, MA, USA). Images were acquired using an Olympus BX40 microscope. Positive and negative cells per field were quantified in a blinded manner using QuPath v0.3.2 [24].

### 2.6. Real-Time Cell Analysis (RTCA) by Electric Impedance Sensing of Invasion and Migration

Tumor cell migration and monocyte invasion were monitored using xCELLigence E- and CIM-Plates (Agilent, Santa Clara, CA, USA) according to the manufacturer’s protocols. Migration assay: A CIM-Plate was used to monitor cell migration across a porous membrane (xCELLigence, Agilent, Santa Clara, CA, USA) using RTCA [27]. HUVEC were plated in the bottom well and treated with 2 µM palbociclib for four hours. MDA-MB-231 cells were then added to the top chamber, which contained a porous membrane and electrode to detect migrating cells. Changes in the electric impedance as cells migrated between chambers was measured at 5 min intervals. Invasion assay: HUVEC (30,000 cells per well) were plated on an E-plate for 24 h until a monolayer was formed. The palbociclib monolayer was then treated with 5 ng/mL recombinant interleukin-1-β (Peprotech, Waltham, MA, USA) and/or varying concentrations of palbociclib for 4 h. THP-1 cells (10,000 per well) were then added to the monolayer to determine their invasive capacity. Graphs show representative data from one of three independent experiments.

### 2.7. RNA Sequencing and GSEA

For murine in vivo experiments, total lung RNA was extracted from the lungs of ATTAC mice treated according to the experimental paradigms above. For in vitro experiments, RNA was extracted from HUVEC that were recovered from cryostorage, passaged once, and then treated with 2 µM palbociclib for 72 h or with vehicle. RNA integrity score (RIN value) for all samples was greater than 8. Sequencing was performed at Novogene Inc. (Sacramento, CA, USA) on an Illumina HiSeq 4000 or NovaSeq 6000 sequencer (paired-end 150 nucleotide reads), resulting in an average of 43 million reads with an average of 93% mapping to the reference genomes. FASTQ files were quality controlled using FastQC (Babraham Institute, Cambridge, UK) [28]. Salmon v1.7.0 [29] was used to align raw sequence reads and estimate expression counts using the GRCm39 mouse genome or the GRCh38.p13 human genome. Subsequently, differential gene expression analyses were performed in R using the edgeR robust generalized linear model pipeline [30]. Gene set enrichment analysis using MSigDB Hallmark or senescence-related gene sets was performed using GSEA 4.2.3 [31,32]. “TGF-β-related” genes were retrieved from the MSigDB Hallmark gene sets. “Leukocyte migration” gene set was retrieved from AmiGO [33,34] under accession number GO:0050900. DAVID enrichment was performed using recommended settings [35].

### 2.8. CIBERSORTx Analyses

CIBERSORTx was used to estimate cell type abundances from bulk RNA sequencing of mouse lungs [36]. A custom cell type signature matrix for mouse lung was first generated in CIBERSORTx using single-cell RNA sequencing data from Smart-seq2 sequencing of FACS-sorted lung cells downloaded from the publicly available *Tabula muris* datasets [37]. Cell fractions were then imputed in CIBERSORTx using recommended settings.

### 2.9. Statistics

Software used for statistical analyses and graphing included R v4.1.2 and Prism v9.5.0 (GraphPad, Inc., La Jolla, CA, USA). Analysis of variance was used for multiple comparisons, and *t*-tests were used for paired comparisons. Unless stated otherwise, values reported indicate mean ± standard error.

### 2.10. Data Availability

RNA sequencing data are available via the Gene Expression Omnibus under accession number GSE223759.

## 3. Results

### 3.1. Palbociclib Increases Seeding of Syngeneic TNBC Cells to the Lung

We used two syngeneic TNBC mouse cell line models to examine the impact of host exposure to treatment with palbociclib, one of the three FDA-approved CDK4/6 inhibitors (CDKis), on metastatic seeding of breast cancer. The E0771 line is a triple-negative breast cancer cell line that forms allograft tumors in C57BL/6 mice and metastasizes to lung and liver from primary tumors or intravenous injection [38,39]. The m6 basal-like breast cancer line is derived from mammary tumors formed in C3(1)-SV40/T-antigen mice [40,41] and grows tumors in C3(1)-SV40/T-antigen-REAR (“REAR”) mice that tolerate tumor cells expressing SV40 T antigen [22]. m6 cells metastasize to the lung and liver from primary tumors or when injected intravenously. The IC_50_ concentrations of palbociclib for the E0771 and m6 lines were greater than 1 µM (Appendix A), defining them as intrinsically resistant to CDKis [42]. Consistent with in vitro resistance, in vivo growth of E0771 or m6 as primary tumors was not altered by treatment of mice with palbociclib (Appendix A). Based on the resistance of these cell lines to palbociclib, we can thus ascribe effects on metastatic spread of the E0771 or m6 lines to the local or systemic drug effects on the host rather than on the cancer cells themselves. To examine the host effects of palbociclib, we treated naïve mice with palbociclib for 10 days at 150 mg/kg and then inoculated them with one million tumor cells via the tail vein, which led to lung seeding of tumor cells (Figure 1A, Appendix A). The lungs of the mice were collected after a monitoring period, and the cross-sectional area of lung metastasis was quantified from histology. Examination of the lungs showed that palbociclib pretreatment significantly increased the percentage of the lung area that was occupied by metastases of E0771 cells from 33.9 ± 9.69% to 64.7 ± 6.73% (*p* < 0.05) (Figure 1B). In the m6 model, similar trends were observed in the percentage area of metastasis and number of metastases per mm^2^ of lung tissue (Appendix A). Taken together, the data from E0771 and m6 intravenous injections in mice indicate that pre-exposure to palbociclib resulted in an increase in the metastatic growth of these mammary cancer cells.

Chemotherapy-induced stromal senescence has previously been shown to increase metastatic spread [15]. Thus, to determine whether the senescence effects of palbociclib pretreatment cause a more hospitable environment for metastatic seeding of circulating mammary cancer cells, we examined the effect of the removal of host senescent cells on E0771 metastatic spread using the p16-INK-ATTAC (“ATTAC”) model [21]. Transgenic ATTAC mice carry a genetic cassette expressed under control of the p16^INK4A^ promoter. p16^INK4A^, transcribed from the *CDKN2A* locus, is a potent inhibitor of the cyclin D-CDK4/6 complex, and its gene expression is frequently activated to drive senescence [18]. p16^INK4A^ activation in cells of ATTAC mice drives transcription of the ATTAC cassette, which produces inactive monomers of caspase-8. Administration of an otherwise inert compound, AP20187 (AP), leads to the dimerization and activation of caspase-8, apoptosis, and depletion of senescent cells. ATTAC mice are generated on a C57BL/6 strain background; thus, these mice tolerate E0771 cells, which are not susceptible to clearance by AP treatment as they do not contain the ATTAC genetic cassette. To determine whether host senescence is involved with the palbociclib-induced enhanced metastatic seeding phenotype, we incorporated into the pretreatment experimental design an additional cohort of mice that received palbociclib and AP (Figure 1C). Palbociclib pretreatment again increased metastasis area on lung cross-section compared to the vehicle group from 2.8 ± 0.6% to 23.8 ± 4.9% (*p* < 0.05) and increased the number of metastatic foci per mm^2^ of lung tissue (from 0.5 ± 0.04 to 2.4 ± 0.2, *p* < 0.01). Interestingly, compared to the palbociclib-only group, addition of AP halved the number of metastatic foci per mm^2^ of lung tissue from 2.4 ± 0.2 to 1.2 ± 0.3 (*p* < 0.05). The combined treatment group also had a trend of smaller lung area occupied by metastasis (11.4 ± 3.7% vs. 23.8 ± 4.9%), although the decrease did not completely return to vehicle-treated baseline levels (Figure 1D,E). We extended these analyses to investigate palbociclib-induced senescence effects on metastasis of primary E0771 tumors. E0771 tumors implanted subcutaneously in the flanks of female ATTAC mice were insensitive to palbociclib, AP, or palbociclib + AP, and growth of recurrent lesions after primary tumor resection was also not different between groups. Interestingly, there was a trend of increased lung metastasis in mice that received palbociclib treatment, and this trend was reversed in the mice that had received palbociclib + AP (Appendix A). This suggests that the p16-related senescence effects of palbociclib in the normal lung contribute significantly to a niche in the lung microenvironment that is more favorable to metastatic seeding.

### 3.2. Gene Expression Changes Accompany Palbociclib-Induced Senescence in Normal Lung

The above pro-metastatic phenotype indicated that palbociclib pretreatment has effects on normal lung tissue, significantly dependent on cellular senescence, that enhance the ability of tumor cells to seed in lungs. Therefore, we next examined the impact of 10 days of palbociclib treatment with or without concomitant AP administration on gene expression patterns in naïve, non-tumor-bearing ATTAC mouse lungs. Previously, in xenograft models, we observed sustained gene expression changes in the tumor-associated stroma after palbociclib withdrawal [19]. To determine whether sustained CDKi-induced gene expression changes could be reversed by continuous AP treatment, RNA sequencing was performed on RNA prepared from the lungs of female ATTAC mice treated with palbociclib, AP, combination palbociclib + AP, or vehicle control. Samples were collected after 10 days of treatment (“Immediate Effect” cohort) or after 10 days of treatment plus 14 days of holiday from palbociclib and continued AP if the initial treatment included AP (“Residual Effect” cohort) (Figure 2A). Differential gene expression analysis comparing each treatment group to the vehicle control of the same timepoint revealed that the naïve ATTAC mouse lung responded to 10 days of palbociclib treatment with profound gene downregulation, whereas the addition of AP to the palbociclib regimen caused both significant up- and downregulation of gene expression (Figure 2B). Most of the highly significant gene changes seen at 10 days in both groups did not persist after 14 days of holiday from palbociclib or after 14 days of continued AP after palbociclib + AP treatment, though many genes were dysregulated in the Residual Effect cohort with large-magnitude fold-change values, albeit to lesser degrees of statistical significance (Figure 2B). AP treatment alone produced few highly significant gene expression changes on the naïve mouse lung, which suggests that senescent cells are rare in naïve ATTAC mice of this age (Appendix A), as we also noted previously in other organs [43]. On the other hand, senescence-dependent gene expression induced after palbociclib treatment became apparent by concomitant administration of palbociclib + AP; comparison between the palbociclib-treated group and the combination-palbociclib + AP-treated group showed that addition of AP negated a large portion of the palbociclib-induced response (Appendix A). Of note, *Lmnb1* and *Foxm1*, which are genes known to be downregulated in senescence [44,45], were also found to be downregulated by 10 days of palbociclib treatment but were not down regulated when senescence reversal with AP occurred in the combination palbociclib + AP group (Figure 2B, Appendix A). Interestingly, combination palbociclib + AP also upregulated 68 genes unique to that treatment group (Appendix A). A smaller portion of the dysregulated genes was either downregulated by both palbociclib and the combination of palbociclib + AP or only downregulated by the combination (Appendix A). In addition to these highly significant gene expression changes, numerous genes were dysregulated in the lungs with large fold-change values but to lesser degrees of statistical significance in both treatment groups and at each timepoint, which is a consequence of the heterogeneity between biological replicates that is captured by bulk RNA sequencing (Figure 2B). To account for this, we used gene set enrichment analysis (GSEA) to resolve the cumulative effects of all gene expression changes and gain insight into the activation or inactivation of biological pathways that affect the lung microenvironment.

Gene set enrichment analysis (GSEA) revealed unique patterns of pathway enrichment in palbociclib- or combination-palbociclib + AP-treated lungs. We found significant positive enrichment in pathways related to cellular senescence, which, interestingly, presented only after 10 days of palbociclib treatment and a drug holiday. Notably, this enrichment in senescence-related gene sets was reversed by coadministration of AP during the palbociclib treatment and continued AP during the palbociclib holiday (Figure 2C). In the MSigDB Hallmark gene sets [31], we found that ten days of palbociclib treatment caused significant negative enrichment in gene sets involved with cell cycle progression, such as E2F targets, G2/M checkpoint, and mitotic spindle, consistent with the mechanism of action of CDKis on the G1/S checkpoint [46] (Figure 2D). In addition to these effects, 10 days of palbociclib treatment induced significant negative gene set enrichment in several pathways related to immune response such as the interferon-γ response, IL2/STAT5 signaling, inflammatory response, IL6/JAK/STAT3 signaling, and complement pathways. No MSigDB Hallmark gene sets were positively enriched to statistical significance after 10 days of palbociclib treatment. We concluded that the immediate and dominant effect of palbociclib treatment on the normal lung is cell cycle and growth arrest that also involves a profound suppressive effect on immune-response-related genes. After 14 days of palbociclib holiday, negative enrichment of cell-cycle-related and immune-response-related gene sets did not persist; however, gene expression related to TGF-β signaling and heme metabolism was elevated, and the myogenesis pathway was found to be suppressed (Figure 2D). Coadministration of AP during the 10-day palbociclib treatment did not reverse palbociclib-induced cell cycle suppression in the lungs, as the G1/S checkpoint and other gene sets expressed after the G1/S checkpoint gene sets remained negatively enriched. However, the addition of AP prevented most negative enrichment in immune-related gene sets seen with 10 days of palbociclib alone. After 10 days of combination palbociclib + AP and a continued 14 days of AP, pathway-level gene dysregulation largely did not persist, and, interestingly, there was no longer a lasting significant enrichment of TGF-β signaling in the lung. This suggests that increased TGF-β signaling is one of the senescence-dependent long-term effects of palbociclib that emerges even after a drug holiday. TGF-β is a component of the senescence-associated secretory phenotype (SASP), and its downstream effects can both enforce a state of cellular senescence by inducing expression of endogenous cyclin-dependent kinase inhibitory proteins and spread paracrine senescence in the local microenvironment [47,48]. Together, this analysis suggests that, with the removal of host senescent cells, the palbociclib-induced negative enrichment of immune pathways is reversed in the lung, and this is accompanied by a profound induction of chemokine and cytokine gene expression. Notably, the elimination of p16^INK4A^-expressing cells does not reduce the cell cycle inhibitory effects of palbociclib on the lung but does attenuate the palbociclib-induced, late-developing enrichment in senescence-related gene sets and TGF-β signaling.

### 3.3. CDK4/6 Inhibition Alters Lung Immune Cell Type Composition

Changes in gene expression patterns in tissues can be due to altered control of gene activation in cells or loss or gain of cell subtypes in tissues, and we conjectured that altered ratios of senescent cell types in the lung treated with palbociclib ± AP could be one cause of the changes we observed. To examine this, we used CIBERSORTx [36] analysis of the bulk RNA sequencing data to determine the drug treatment effects on overall composition of cell types in the normal mouse lung. This software deconvolves bulk RNA sequencing reads from a mixed tissue sample to give a readout of relative composition of cell types. We used a publicly available normal tissue reference atlas of single-cell RNA sequencing data from *Tabula muris* [37] to generate a cell type gene expression signature matrix for the normal mouse lung. Using this reference matrix in CIBERSORTx, we estimated the abundance of the reference cell types in our samples. Despite the profound inhibitory effects of palbociclib on cell cycle gene expression in the lungs, we did not see changes in the abundance of the major lung cell types; regardless of treatment group, the samples were mostly composed of endothelial (31.3 ± 0.5%), epithelial/tracheobronchial (23.2 ± 0.3%), stromal (20.8 ± 0.2%), and neuroendocrine (19.5 ± 0.2%) cells (averages reported across all treatment groups) (Figure 3A). The remaining portion (<10%) of each sample was comprised of leukocytes, and analysis of these subpopulations revealed significant changes in the macrophage/monocyte cell types (Figure 3A, right panel). After 10 days of palbociclib treatment, there was a significant decrease in the overall macrophage/monocyte levels versus vehicle control from 5.9 ± 0.4% to 2.1 ± 0.05% (*p* < 0.05). AP treatment significantly reversed some of the palbociclib-induced losses in total macrophage and monocyte levels from 2.1 ± 0.05% to 3.7 ± 0.2% (*p* < 0.05), though overall levels in the combination group were still below the vehicle baseline (Figure 3B). Macrophage and monocyte subtypes were also significantly altered in the palbociclib-treated group. Levels of dendritic cells, alveolar macrophages, and interstitial macrophages were decreased by palbociclib treatment versus vehicle control from 3.0 ± 0.2% to 0.6 ± 0.2% (*p* < 0.01), circulating monocytes were decreased by palbociclib treatment from 2.9 ± 0.6% to 0.9 ± 0.1% (*p* < 0.05), and there was an induction of invading monocytes by palbociclib treatment from levels below detection to 0.7 ± 0.3% (*p* < 0.05). Addition of AP also reversed the trends of palbociclib-induced decreases in macrophage subtypes and invading monocytes, albeit not to the vehicle baseline. Adding AP did not alter the palbociclib-induced appearance of invading monocytes, which remained elevated in the palbociclib + AP group versus vehicle control from a level below detection to 0.8 ± 0.1% (*p* < 0.01) (Figure 3C).

Given the palbociclib-induced alterations to immune cell abundance in the lung and the ability of concomitant treatment with AP to prevent some of these changes, we reasoned that palbociclib-induced cellular senescence in the lung may affect gene expression that modulates macrophage and monocyte recruitment to this metastatic niche. We examined differential expression in the lung of genes associated with the gene ontology (GO) term “leukocyte migration” retrieved from AmiGO [33,34] (accession number GO:0050900) to assess for palbociclib- or palbociclib + AP-induced changes (Figure 3D). Of the murine genes included in this annotation, 17 were significantly downregulated by palbociclib: *Ccl5*, *Ccl17*, *Ccr7*, *Coro1a*, *Cxcr3*, *Cxcr5*, *Gcsam*, *Gpr183*, *Il16*, *Itgb7*, *Jaml*, *Prex1*, *Ptpn22*, *Sell*, *Slamf9*, *Spn*, and *Tbx21*. Of these, only *Ccl5*, *Gcsam*, *Itgb7*, and *Jaml* were also downregulated by palbociclib + AP, indicating that palbociclib-induced downregulation of most of these genes was reversed by AP. There were no genes in the “leukocyte migration” annotation that were significantly upregulated by palbociclib alone; however, 11 genes were significantly upregulated by palbociclib + AP: *Bdkrb1*, *Ccl3*, *Ccl6*, *Cxcl1, Cxcl2*, *Cxcl5, Cxcr1*, *Cyp7b1*, *Ror2*, *Spp1*, and *Trem2*. Of note, this subset of upregulated genes seen in the lungs treated with palbociclib + AP is likely to be related to influx of neutrophils and an innate immune response mounted against apoptotic senescent cells (Appendix A). Taken together, our data indicate that palbociclib causes multiple cell-cycle- and immune-related changes in the normal lung without changing overall cell composition. AP-induced removal of senescent cells reverses a subset of these changes but also induces new gene upregulation mainly related to neutrophil activation and innate immunity. This occurs in parallel to the changes in macrophage cell populations caused by palbociclib that are partially reversed by AP treatment.

### 3.4. Senescent Endothelium Contributes to Pro-Metastatic Phenotype

The effects of palbociclib on lung macrophages could be acting through an autocrine or paracrine senescence effect that has been described previously [47,48]. To examine which lung stromal cells might be the most sensitive to palbociclib-induced senescence effects, we examined with immunofluorescence (IF) the expression of the ATTAC cassette using its intrinsic GFP tag. Unfortunately, the background fluorescence in the lung parenchyma was too high to resolve GFP expression to individual cells; however, the lower background within the lung metastases in ATTAC mice permitted examination of which cell types had predominant GFP staining. The lung is highly vascularized, and we examined. the contribution of the endothelium to the lung stromal responses to palbociclib using IF staining for CD31 to identify endothelial cells in palbociclib-pretreated E0771 metastases in ATTAC mice. We overlaid this analysis with IF staining for expression of the ATTAC cassette using its intrinsic GFP tag to identify the p16^INK4A^-expressing senescent cells. Strikingly, there was a portion of endothelial cells within the lung metastases that stained positive for expression of the ATTAC cassette (Figure 4A). To determine whether the effects of palbociclib could impact directly on endothelial cells, we treated human umbilical vein endothelial cells (HUVEC), which are primary cells derived from the human vascular endothelium, with palbociclib in vitro. We found that HUVEC had a biphasic dose response to palbociclib, with a separate IC_50_ to palbociclib of 261.8 nM and 14.4 µM, suggesting that palbociclib treatment elicits different responses related to cell survival in HUVEC through different pathways (Appendix A). Using 72 h of palbociclib treatment with a dose of 2 µM, which is between both IC_50_ values, we observed that, compared to control HUVEC, the majority of palbociclib-treated HUVEC readily displayed hallmark features of cellular senescence such as increased staining for senescence-associated β-galactosidase (SA-β-gal), G1-phase cell cycle arrest, and increased cell size when grown in vitro (Figure 4B–D, Appendix A). Of note, the percentage of HUVEC stained positive for SA-β-gal after palbociclib treatment is comparable to levels of SA-β-gal staining seen in HUVEC after doxorubicin treatment or irradiation [49]. Treatment of HUVEC with abemaciclib, another CDK4/6 inhibitor [50], also induced a significant increase in SA-β-gal positivity and G1-phase cell cycle arrest in HUVEC, which indicates that these effects are largely due to inhibition of the cyclin D-CDK4/6-RB pathway (Appendix A).

Cellular senescence is known to be accompanied by distinct gene expression changes [14]. To examine whether similar effects could be seen in HUVEC, we performed RNA sequencing on palbociclib-treated HUVEC. We found that palbociclib induced profound gene dysregulation in HUVEC (Figure 4E). Some of the most significantly downregulated genes in HUVEC treated with palbociclib included *KIF20A*, *MKI67*, *MYBL2*, *RRM2*, and *TOP2A*, which are associated with cell cycle progression and lost in growth arrest. We also found downregulation of *FOXM1* and *LMNB1*, which are signature genes known to be downregulated in senescence [44,45]. We probed common endogenous CDK-inhibitory genes and found significant upregulation of *CDKN1A* and *CDKN2B*, which are TGF-β pathway target genes that encode the endogenous cyclin-dependent kinase proteins and potent enforcers of cytostasis p21^CIP1^ and p15^INK4B^. In fact, several TGF-β-signaling-related genes were also significantly dysregulated in HUVEC treated with palbociclib, including *TGFB1*, which encodes TGF-β, and could create positive feedback that reinforced paracrine senescence (Figure 4F). This is consistent with previous reports that the activity of TGF-β can be enhanced by CDKis. One of the additional physiological roles of CDK4/6 is to phosphorylate and inactivate SMAD2, which is a downstream effector protein in the TGF-β signal transduction cascade. Palbociclib inactivation of CDK4/6 prevents SMAD2 inactivation, and, thereby, enhances TGF-β signaling activity [51,52]. The profound cytostasis that occurs as a combination of CDK4/6 inhibition and parallel TGF-β pathway activation of p21^CIP1^/p15^INK4B^ induced by palbociclib could result in production of the senescence-associated secretory phenotype (SASP). Palbociclib-treated HUVEC demonstrated upregulation of a number of SASP-associated genes (Appendix A) [14]. We then returned to examine genes associated with leukocyte migration to assess the ability of senescent endothelial cells to affect leukocyte behavior. One of the most highly upregulated genes in HUVEC by palbociclib treatment was *VCAM-1*, which is a key mediator of leukocyte rolling and adhesion to the endothelium during extravasation (Figure 4E) [53]. Additionally, of the human genes in the gene ontology annotation “leukocyte migration” retrieved from AmiGO, 39 were significantly upregulated in HUVEC by palbociclib treatment (Figure 4F). This is consistent with the well-documented feature of cellular senescence whereby it induces pro-migratory and inflammatory factors that increase immune cell motility [14]. Gene set enrichment analysis of the palbociclib-treated HUVEC confirmed that the downregulated genes were predominantly related to cell division and showed that senescent endothelial cells participate in a producing a proinflammatory milieu, as there was enrichment of immune response-related gene sets including interferon-α/γ response, IL6-JAK/STAT signaling, and inflammatory response (Figure 4G). We concluded that endothelial cells exposed to palbociclib exhibit a phenotype that recapitulates many hallmark features of cellular senescence and involves gene expression pattern changes that increase inflammatory and leukocyte migration signaling.

To investigate the potential cross-talk between senescent endothelial cells and resident host cells or tumor cells in the metastatic niche, we performed co-culture assays using palbociclib-treated HUVEC and monocyte or breast cancer cell lines. A three-chamber co-culture system using palbociclib-treated HUVEC and serum-starved MDA-MB-231 breast cancer cells was assembled to assess the ability of secreted factors from senescent HUVEC to induce migration of breast cancer cells. In this assay, two co-cultured cell types were separated by a semi-permeable membrane that permitted movement of soluble secreted factors but not cells and a second membrane to measure migration (Figure 5A) [27]. MDA-MB-231 cells were cultured in serum-free DMEM for 6 h, while HUVEC were plated in the bottom chamber of the co-culture system and treated with 2 µM palbociclib for 4 h. MDA-MB-231 cells were then added to the upper chamber, and their migration in the direction of the HUVEC was measured by electric impedance sensing with real-time cell analysis (RTCA) [27]. After 8 h of co-culture, the MDA-MB-231 cells that were co-cultured with palbociclib-treated HUVEC had migrated in the direction of the endothelial cells to a greater degree than the MDA-MB-231 cells that were co-cultured with vehicle-treated HUVEC, indicating that palbociclib treatment of HUVEC was able to increase the baseline migration level of serum-starved MDA-MB-231 cells (Figure 5B). Next, we examined the functional consequences of palbociclib in a model of trans-endothelial invasion using direct-contact co-culture of HUVEC and THP-1 cells. The THP-1 line is a human leukemia monocyte cell line commonly used to study monocyte and macrophage function and is palbociclib resistant [54,55]. In this assay, HUVEC were plated at high density on an RTCA electrode and grown to a confluent monolayer overnight [56]. The monolayer was then treated with 0.2 µM palbociclib, 2 µM palbociclib, 5 ng/mL IL-1β, or combination 2 µM palbociclib + 5 ng/mL IL-1β for four hours. IL-1β is used as a positive control to measure the maximum possible invasion as it is an endogenous activator of cellular extravasation [57]. Subsequently, THP-1 cells were added to the well, and their disruption of the HUVEC monolayer was measured by RTCA (Figure 5C). As expected, we found that 5 ng/mL IL-1β treatment of the HUVEC monolayer significantly increased THP-1 trans-endothelial invasion (Figure 5D). Additionally, 2 µM palbociclib treatment also significantly increased THP-1 trans-endothelial invasion of the HUVEC monolayer versus vehicle treatment as percent invasion relative to the IL-1β control (100%) from 21.8 ± 5.7% to 52.6 ± 1.6% (*p* < 0.01; Figure 5E). Collectively, these data indicate a broad effect of palbociclib on normal tissues in the lung, including increased endothelial senescence, as well as redistribution of macrophages and monocytes, which can contribute to preparing the metastatic niche for seeding of cancer cells.

## 4. Discussion

The advent of CDK4/6 inhibitors has been a major milestone in the treatment of metastatic breast cancer, but long-term efficacy of these drugs is limited by intrinsic and acquired therapeutic resistance. Here, we assessed the impact of CDKi-induced cellular senescence on metastatic seeding and metastatic growth of breast cancer. We found that palbociclib pretreatment increased seeding of CDKi-resistant syngeneic mammary cancer cell lines in the lung. By using the p16-INK-ATTAC transgenic mouse model of inducible senolysis, we showed that palbociclib-enhanced lung seeding can be partially mitigated by removal of p16^INK4A^-expressing host senescent cells. Our findings are also consistent with reports using other syngeneic mouse models of breast cancer that palbociclib pretreatment can enhance lung seeding of breast cancer cells after intravenous injection through a mechanism that correlates with stromal senescence [58]. In addition, host senescence previously has been implicated in enhanced metastatic spread of syngeneic breast cancer cells. Selective removal of host senescent cells mitigated this effect in a series of experiments conducted using doxorubicin-induced senescence in p16-3MR mice, a similar murine model of inducible senolysis of p16^INK4A^-expressing cells [15]. Doxorubicin is a potent inducer of cellular senescence both in vitro and in vivo, but, unlike CDKis, it causes significant DNA damage and generation of free radicals [15,59,60]. The data presented here argue that the increased metastatic seeding of CDKi-resistant mammary cancer cells occurs as a common consequence of therapy-induced stromal senescence regardless of the mode of induction. This phenotype is of particular importance in the context of drug-resistant disease; while the benefits of CDKis in relation to progression-free survival are well established in patients with widespread metastatic disease [9], therapeutic manipulation of host responses to CDKis offers an additional approach toward disease management in the context of acquired or intrinsic resistance.

We identified senescent endothelial cells within the lung metastases of palbociclib-pretreated mice and found that endothelial cells in culture readily displayed hallmark features of cellular senescence when treated with palbociclib. Primary endothelial cells treated with palbociclib demonstrated cell cycle arrest, increased staining for senescence-associated β-galactosidase, and upregulation of genes involved with production of the senescence-associated secretory phenotype (SASP) as well as several immune-signaling-related pathways. Among one of the most highly upregulated genes in human umbilical vein endothelial cells is *VCAM-1*, which encodes a cell adhesion protein that mediates rolling and adhesion of circulating leukocytes to facilitate extravasation during inflammation [53,57]. Palbociclib-induced expression of *VCAM-1*, as well as upregulation in HUVEC of several other genes involved in leukocyte migration, could be responsible for palbociclib treatment inducing the pro-invasive effect we observed for THP-1 trans-endothelial invasion in vitro (Figure 5). This is consistent with the palbociclib-induced increase in invading monocytes seen in our in vivo analyses using CIBERSORTx (Figure 3).

Additionally, our data show that TGF-β-signaling-related gene enrichment in the lungs is a residual effect of palbociclib, which was revealed after a 14-day drug holiday, and correlates with positive enrichment of senescence-related gene sets in the lungs and is dependent on the presence of p16^INK4A^-expressing senescent cells. One of the main stromal effects of the TGF-β pathway is the achievement of cytostasis by inducing expression of endogenous cyclin-dependent kinase inhibitory proteins such as p21^CIP1^ and p15^INK4B^ and by suppressing proliferative proteins such as c-MYC [61]. Furthermore, development of senescence has been shown to be dependent on intact TGF-β signaling. In cell line models of oncogene-induced senescence, silencing TGF-β receptor type-2 activity was sufficient to suppress senescence induction in breast fibroblasts [62]. TGF-β is also secreted as part of the SASP and can induce paracrine senescence in the surrounding microenvironment [47]. Paracrine senescence was shown to be also dependent on intact TGF-β activity as its development can be blocked by TGF-β-neutralizing antibodies [63]. SMAD2 and SMAD3 are two highly similar proteins that act as the main effectors of canonical TGF-β signal transduction. Upon TGF-β receptor activation, SMAD2/3 is activated by phosphorylation and translocates to the nucleus, where it complexes with SMAD4 and other cofactors to drive transcription of TGF-β pathway target genes [61]. CDK4 is a key negative regulator of TGF-β signaling, as this enzyme has been shown to phosphorylate and inactivate SMAD2/3 [52,64]. Intriguingly, palbociclib interacts with the TGF-β pathway through blocking CDK4-mediated inactivation of SMAD2; in T47D breast cancer cells, palbociclib enhanced SMAD2 binding to the genome by reducing its inhibitory phosphorylation by CDK4/6, resulting in enhanced expression of TGF-β target genes *CDKN2B* and *PMEPA1* [51]. In our hands, HUVEC treated with palbociclib also displayed significant upregulation of both *CDKN2B* and *PMEPA1*, as well as several other TGF-β target genes including *CDKN1A* and *CDKN1C* (Figure 4), adding increased downstream TGF-β pathway activation to the scope of palbociclib-induced changes in human endothelial cells. Additionally, in a different study from our lab using the p16-INK-ATTAC model, we observed endothelial senescence in kidneys after angiotensin II treatment of animals and its reversal after co-treatment with AP. This was paralleled by upregulation of both *Cdkn1a* and *Cdkn2B* in the kidneys, which was reversed by AP co-treatment [43]. It is noteworthy that these changes were confined to the kidneys, and no senescence signals were observed in the lungs.

Of note, we identified p16^INK4A^-expressing endothelial cells in lung metastases of palbociclib-pretreated p16-INK-ATTAC mice; however, *CDKN2A*, encoding p16^INK4A^, was not differentially regulated by palbociclib in our in vitro analyses of primary endothelial cell lines. Given the significant palbociclib-induced upregulation of *CDKN1A* and *CDKN2C* that we observed in HUVEC and the relation of p21^CIP1^ and p15^INK4B^ to paracrine senescence by way of TGF-β signaling, it is possible that a cellular subpopulation sensitive to palbociclib and acting through a paracrine mechanism could be responsible for inducing p16^INK4A^ expression in endothelial cells within the metastasis, thus, modulating the propensity for the lung to support metastatic growth and the invading monocyte populations. In models of acetaminophen-induced liver injury, small-molecule inhibitors of TGF-β receptor type 1 improved survival and enhanced tissue recovery after acute hepatotoxicity by blocking TGF-β-induced paracrine senescence [65]. Future studies into cellular subpopulations responsible for palbociclib-induced paracrine senescence will be greatly benefitted by resolving gene expression changes in the CDKi-pretreated lung at the single-cell level to probe cellular subtype-specific effects and by incorporating other methods of inducible senolysis to fully probe the relationship between p16^INK4A^- and p21^CIP1^-mediated cellular responses after palbociclib treatment. Mouse models that permit inducible removal of p21^CIP1^-expressing host senescent cells [17], as well as investigation using TGFβ receptor inhibition and clinically relevant senolytics, can be leveraged for this purpose. One such clinically relevant senolytic is venetoclax, which is a small-molecule inhibitor of BCL2 used in the treatment of chronic lymphocytic leukemia and other hematologic malignancies [66,67]. In patient-derived xenograft models, triple therapy with venetoclax, palbociclib, and fulvestrant attenuated tumor growth and resulted in superior survival versus monotherapy or doublet combinations, and triple therapy with palbociclib, letrozole, and venetoclax is currently under investigation in the PALVEN phase 1b clinical trial (NCT03900884) [68,69].

## 5. Conclusions

In conclusion, we found that stromal senescence induced after palbociclib treatment increased metastatic seeding and growth of CDK4/6-inhibitor-resistant breast cancer cells. Palbociclib-induced negative enrichment of immune-related gene sets, suppression of gene expression related to leukocyte migration, and delayed enrichment in TGF-β- and senescence-related gene sets in the lungs of treated naïve mice were mitigated by removal of p16^INK4A^-expressing host senescent cells. Senescence in the metastatic microenvironment, including in endothelial cells, resulted in an influx of invading monocytes. Palbociclib-treated primary endothelial cells in culture exhibited hallmark senescent features. These studies shed light on how stromal senescence induced by palbociclib affects lung metastasis seeding and provide a foundation for mechanistic inquiry into CDK4/6-inhibitor-induced cellular senescence in the future using additional transgenic and pharmacological models of inducible senolysis.

## Figures and Tables

**Figure 1 cancers-15-01908-f001:**
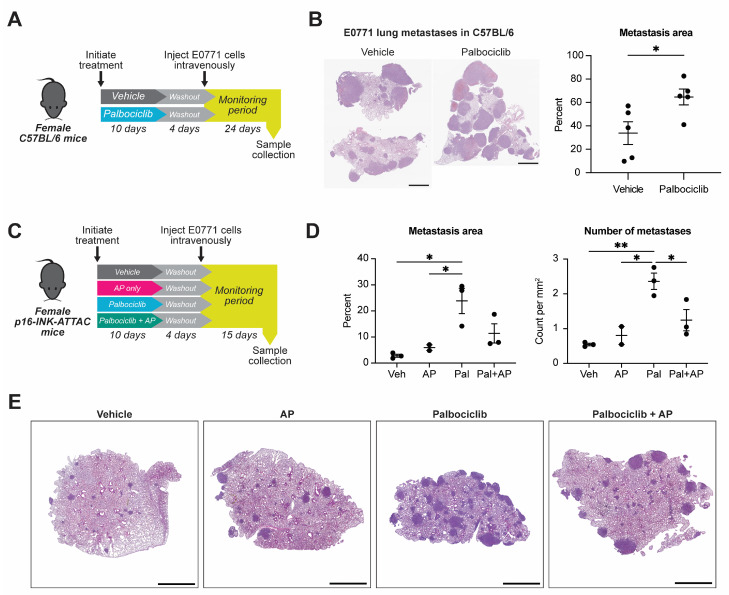
Elimination of senescent host stromal cells reduces palbociclib-enhanced seeding of E0771 TNBC cells in the mouse lung. (**A**) Experimental schematic: Female C57BL/6 mice received 10 days of 150 mg/kg palbociclib daily by oral gavage (10 doses) or vehicle, followed by a 96 h drug washout period and intravenous injection of 10^6^ E0771 cells in the tail vein. Mice were then monitored for 24 days and euthanized. (**B**) Representative H&E-stained lung sections and quantification of cross-sectional lung area of histological section occupied by E0771 metastases from the design described in (**A**). *n* = 5 mice per group; mean ± standard error indicated. * *p* < 0.05 by unpaired *t*-test; scale bars 2 mm. (**C**) Experimental schematic: Female ATTAC mice received 10 days of 150 mg/kg palbociclib daily by oral gavage (10 total doses), AP20187 (AP) 2 mg/kg every three days by intraperitoneal injection (four total doses), and both palbociclib and AP or vehicle control throughout, followed by a 96 h drug washout period and injection of 10^6^ E0771 cells intravenously in the tail vein. Following injection, mice were monitored for 15 days and then euthanized. (**D**) Quantification of E0771 metastases as a percentage of cross-sectional lung area on histological section and as a count of metastatic foci per mm^2^ of lung area. Veh = vehicle (*n* = 3 mice), AP = AP20187 (*n* = 2 mice), Pal = palbociclib (*n* = 3 mice), Pal + AP = combination treatment (*n* = 3 mice); compared by one-way ANOVA with Tukey’s multiple comparisons test: * *p* < 0.05; ** *p* < 0.01. (**E**) Representative H&E-stained lung sections from experiment described in (**C**); scale bars 2 mm.

**Figure 2 cancers-15-01908-f002:**
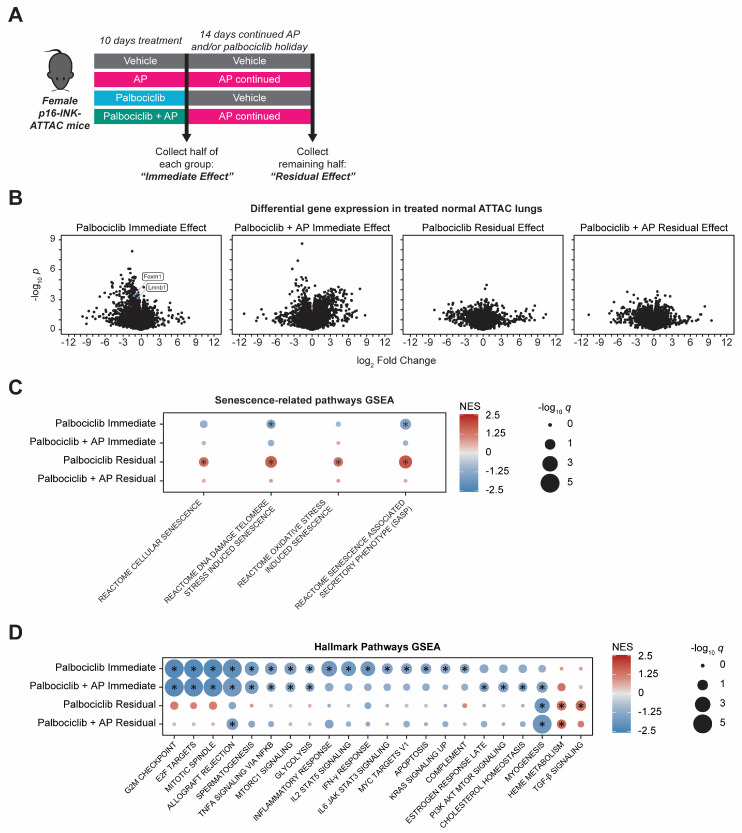
Palbociclib-induced gene expression changes in normal lung are partially reversed by coadministration of AP20187. (**A**) A cohort of 24 female ATTAC mice were randomized into four groups (*n* = 6 per group) for a 10-day treatment period: vehicle, AP20187 (AP) 2 mg/kg every three days (four total doses) by intraperitoneal injection, palbociclib 150 mg/kg daily by oral gavage (10 total doses), or combined palbociclib + AP. Following 10 days of treatment, three mice in each of the groups were euthanized for analysis (“Immediate Effect” cohort). The remaining mice were treated with a 14-day holiday from palbociclib or with 14 days of continued AP 2 mg/kg every three days (five additional doses) prior to collection (“Residual Effect” cohort). (**B**) Volcano plots of differentially expressed genes (DEGs) regulated by 10 days of palbociclib with or without drug holiday or by 10 days of palbociclib + AP with or without 14 days of continued AP. Three independent biological replicates (one RNA sample from each mouse) sequenced per group. Full list of DEGs available in Appendix A. (**C**,**D**) Gene set enrichment analysis of lung RNA from experiment described in (**A**), for senescence-related pathways (**C**), and for MSigDB Hallmark gene sets (**D**). Color indicates normalized enrichment score (NES), and size indicates statistical significance. All pathways graphed reached statistical significance in at least one treatment condition or were excluded: * absolute value of normalized enrichment score (NES) > 1.25 and *q* < 0.25.

**Figure 3 cancers-15-01908-f003:**
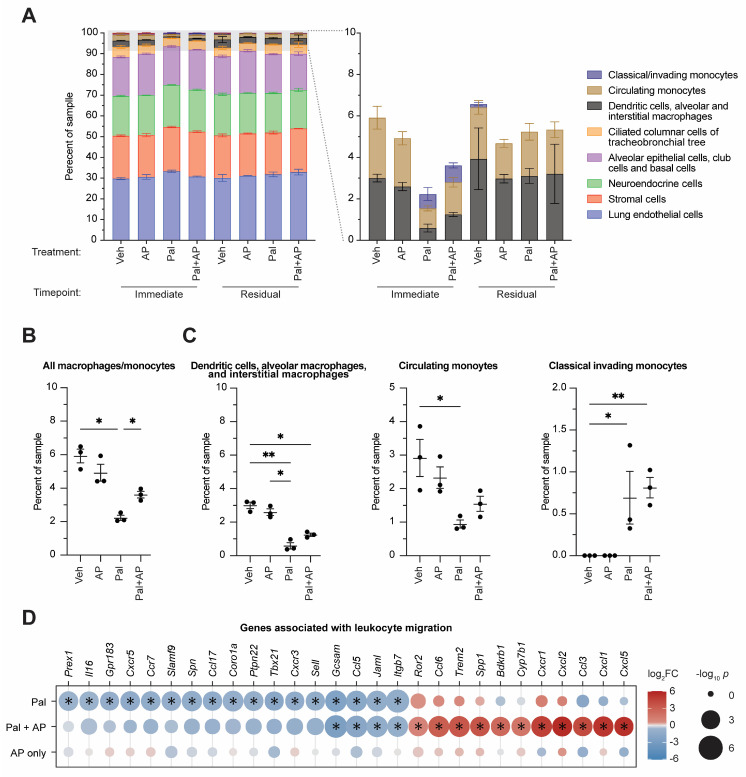
Palbociclib and senescence reversal alter macrophage and monocyte composition in normal mouse lungs. (**A**) CIBERSORTx analysis of mRNA from lungs in different treatment groups treated with vehicle (Veh), AP20187 (AP), palbociclib (Pal), or palbociclib and AP combination (Pal + AP) collected immediately after 10 days of treatment (“Immediate Effect” cohort) or after 10 days of treatment plus 14 days of palbociclib holiday or continued AP (“Residual Effect” cohort). (**B**) Sum of macrophage and monocyte cell types as a percentage of each sample. (**C**) Macrophage or monocyte subtypes as a percentage of each sample. Zeros indicate measurements that were below detection threshold. (**B**,**C**) * *p* < 0.05, ** *p* < 0.01 by one-way ANOVA with Tukey’s multiple comparisons test. (**D**) Differentially expressed genes in treated ATTAC lungs from the gene ontology annotation “leukocyte migration” retrieved from AmiGO under accession number GO:0050900. Color indicates differential expression; size indicates significance level. * differential gene expression surpassing statistical significance threshold of absolute value of log_2_ fold change > log_2_(1.5) and *p* < 10^−3^. Differentially expressed genes graphed reached statistical significance threshold or were excluded.

**Figure 4 cancers-15-01908-f004:**
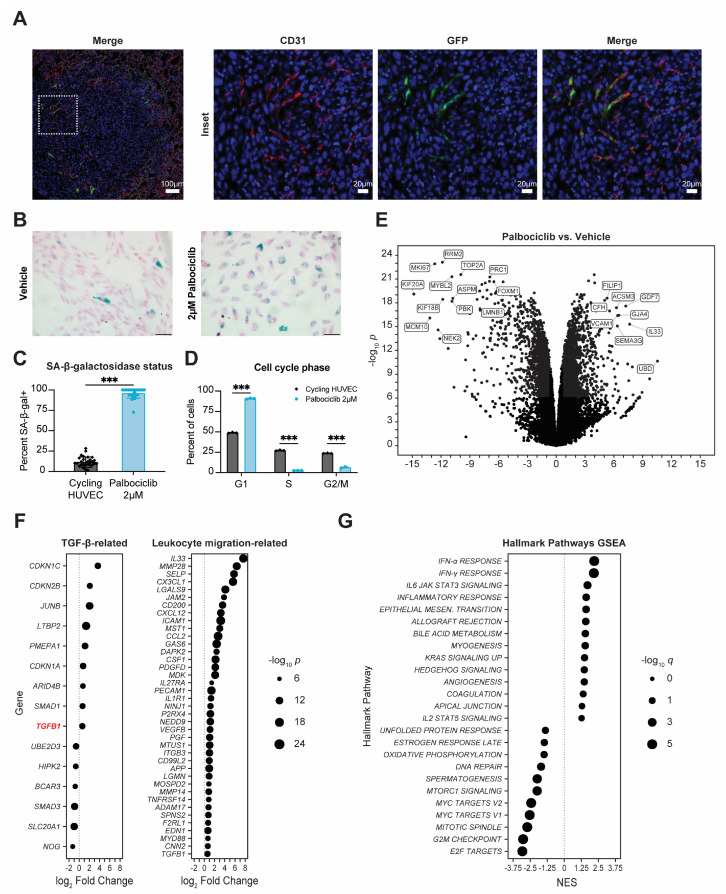
Palbociclib-treated endothelial cells demonstrate hallmark features of cellular senescence in vitro. (**A**) Representative immune fluorescence staining of E0771 metastases in palbociclib-pretreated lungs. Red = CD31; green = GFP; blue = Hoechst dye; scale bars = 100 µm (overview image) or 20 µm (insets). (**B**) Senescence-associated β-galactosidase expression status of HUVEC treated with 2 µM palbociclib versus cycling HUVEC. Nuclear Fast Red counterstain with 50 µm scale bar. (**C**) Quantification from (**B**): thirty non-overlapping 20× fields captured per dose compared by unpaired *t*-test. (**D**) Cell cycle status of HUVEC treated with palbociclib for 72 h compared to cycling HUVEC, quantified by propidium iodide stain and flow cytometry. Three independent replicates per treatment condition compared by two-way ANOVA with Tukey’s multiple comparisons test. (**C**,**D**) All bars indicate mean ± standard error; *** *p* < 0.001. (**E**) Differentially expressed genes in HUVEC induced by 2 µM palbociclib treatment for 72 h. Full list of differentially expressed genes in Appendix A. (**F**) Differentially expressed genes in palbociclib-treated HUVEC from the MSigDB Hallmarks TGF-β signaling gene set or from the gene ontology annotation “leukocyte migration” retrieved from AmiGO under accession number GO:0050900. Differentially expressed genes graphed reached statistical significance threshold of absolute value of log_2_ fold change > log_2_(1.5) and *p* < 10^−6^ or were excluded. Size indicates significance level. (**G**) Gene set enrichment analysis (GSEA) of human MSigDB Hallmark gene sets associated with palbociclib treatment of HUVEC. Pathways graphed reached statistical significance threshold of absolute value of normalized enrichment score (NES) > 1.25 and *q* < 0.05 or were excluded. Size indicates significance level.

**Figure 5 cancers-15-01908-f005:**
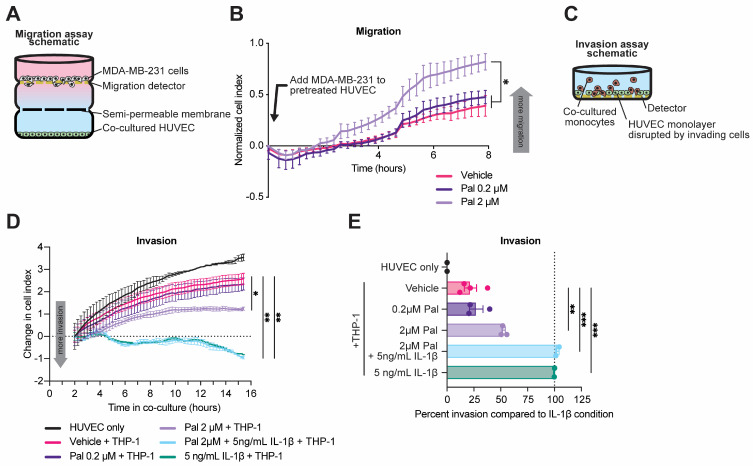
Palbociclib treatment of HUVEC modulates tumor cell migration and monocyte invasion in vitro. (**A**) Schematic of three-chamber co-culture system. (**B**) MDA-MB-231 migration after initiation of co-culture. HUVEC treated with vehicle or palbociclib (Pal) at indicated doses. Increasing cell index indicates more migration. (**C**) Schematic of trans-endothelial invasion co-culture assay. (**D**) Trans-endothelial invasion of THP-1 monocytes exposed to pretreated endothelial cell monolayer with indicated treatments of vehicle, palbociclib, or recombinant interleukin-1β (IL-1β) at indicated doses. Change in cell index is reported beginning two hours after initiation of co-culture. (**B**,**D**) compared by two-way repeated measures ANOVA with Dunnett’s multiple comparisons test, * *p* < 0.05, ** *p* < 0.01. (**E**) Measurements of THP-1 invasion relative to IL-1β positive control and non-THP-1-containing condition as negative control compared by one-way ANOVA with Tukey’s multiple comparisons test. ** *p* < 0.01, *** *p* < 0.001.

## Data Availability

RNA sequencing data is available on the Gene Expression Omnibus under accession number GSE223759.

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
