# Peer review of "Stromal Senescence following Treatment with the CDK4/6 Inhibitor Palbociclib Alters the Lung Metastatic Niche and Increases Metastasis of Drug-Resistant Mammary Cancer Cells"

_cancers, 2023, doi:10.3390/cancers15061908_

Round 1

Reviewer 1 Report

The manuscript "Stromal senescence following treatment with the CDK4/6 inhibitor palbociclib alters the lung metastatic niche and increases metastasis of drug-resistant mammary cancer cells" by Gallanis et al., focuses on elucidating the fundamental mechanisms underlying palbociclib (a CDK4/6 inhibitor) induced treatment resistance in majority of metastatic breast cancer (MBC) patients. CDK4/6 inhibitors along with endocrine therapies are commonly used as 1st line therapy in MBC. However, treatment response is suboptimal. There is a critical knowledge gap in terms of de novo or acquired resistance in patients to palbociclib. The premise of the manuscript is significant and it offers to bridge some aspects of this knowledge gap.

The documentation and the results are compelling. The fundamental strength of the manuscript is authors access to the transgenic p16-INK-ATTAC (“ATTAC”) mice and other critical lines. The authors provided both in vivo and in vitro results to drive home the fundamental message, i.e. palbociclib resistance induces pre metastatic niche in lungs primarily. The authors demonstrated that blocking p-16 accumulation prevents metastatic niche formation in lungs and concurrently there are up and down regulation of myriad of genes in palbo, palbo+ AP and palbo drug holiday+ only AP treated cohorts compared to control groups. The statistical analysis and explanations are quite convincing. The authors use two palbociclib resistant mouse models and that is convincing. 

However, in vivo experiments that are pertinent to address metastatic changes, it's recommended to excise the primary tumors first to ascertain metastatic changes in order to emulate human disease more accurately. Additionally, matrix metalloproteases (MMPs) induction is a hallmark in pre metastatic niche. The authors should examine the induction of MMP9 in metastatic lesion.

This manuscript is clearly written and rationale clearly defined barring a few critical experiments. The addition of these critical experiments will strengthen the manuscript significantly.

Author Response

We thank the Reviewers for their time and effort in critiquing our manuscript. We appreciate their thoughtful commentary and suggestions for improvement of the text and have responded to their critiques as described below.

"However, in vivo experiments that are pertinent to address metastatic changes, it's recommended to excise the primary tumors first to ascertain metastatic changes in order to emulate human disease more accurately."

We have included new data as Supplemental figure 1B, which has data from an experiment in which primary E0771 tumors were treated with palbociclib, AP20187, or palbociclib + AP20187, then resected, allowed to regrow, and then the mice were collected. We show that neither the primary nor recurrent tumors are sensitive to any of the drug combinations and that though the tumor burden is not statistically significantly different between treatment groups, there is a trend of increased lung metastasis in the palbociclib-treated mice and a reverse trend in the palbociclib + AP treated mice. The metastasis data is added in Supplemental figure 1G and we have updated the Results text accordingly.

"Additionally, matrix metalloproteases (MMPs) induction is a hallmark in pre metastatic niche. The authors should examine the induction of MMP9 in metastatic lesion."

We examined MMP9 differential expression in palbociclib or palbociclib + AP treated ATTAC lungs and found that on its own it is not a significantly dysregulated gene. However, MMP9 is one of the genes in the MSigDB Hallmarks gene sets that were used for gene set enrichment analysis (GSEA): Allograft Rejection, Apical Junction, Coagulation, and KRAS Signaling Up. These gene sets were all significantly upregulated in the GSEA of palbociclib-treated HUVEC presented in Figure 4.

Reviewer 2 Report

This is a very extensive and a very interesting study that reveals the role of stromal-niche senescence induced by a major drug (CDK4/6 inhibitor palbociclib) in promoting tumor engraftment in the lung metastatic niche. The experiments seem to be done at an excellent technical level and exhaustively analyzed. However, the story, as presented, is rather confusing and even appears at times to be self-contradictory, which raises unanswered questions and detracts from the value of this very important work. I would strongly recommend revising and restructuring the paper, as discussed below.

Specifically, Figure 1 carries the main message of the paper that palbo pretreatment stimulates lung metastasis via the induction of senescent cells. However, subsequent analysis of the transcriptomic effects of palbo on tumor-free lungs (Figures 2-3) appears to contradict the picture that emerges from the data in Fig. 1.  There seems to be no evidence that palbo induces senescence or SASP in the lungs. Strikingly, palbo apparently does not induce the expression of any genes, the pre-requisite for SASP. In fact, it appears that AP senolysis, which should have attenuated the paracrine effects of senescence, by itself induces gene expression, including chemoikines and cytokines, apparently the opposite to what’s expected. A tenuous link to the overall concept comes from TGF-b pathway observations, but it is not clear if TGF-b effects involve anything that can be biologically linked to the promotion of metastases.

The next part of the analysis of lung RNA-Seq comes from the detailed discussion of the effects on the immune cell composition, based on CIBERSORTx analysis of bulk RNA-Seq data. I have no direct experience with this analysis strategy and I would have felt more comfortable about it if the conclusions about changes in the immune cell subtypes could be verified by immunostatining or FACS analysis, which they weren’t. So: does palbo pretreatment decrease the immune response, which could have accounted for the promotion of metastasis, even if it were senescence independent?

However, the endothelial cell data in Figs. 4-5 brings us back to the concept of Fig. 1 and appears to fit it very well: induction of senescence and SASP, which is attenuated by senolysis. Furthermore, this part of the story feels more biologically relevant, as it starts from the observations in the actual lung metastases and not in tumor-free lungs (in contrast to Figs. 2-3) and it addresses cells where ATTAC cassette is activated. Could it be that clear results in the endothelial cells, as opposed to obscure conclusions from the lungs could reflect the additional component of tumor-stromal interaction, beyond palbo pretreatment?

With this in mind, I would suggest two ways of revising this MS. The first one is to remove Figs. 2-3 altogether, towards a separate paper, since the endothelium was the only identifiable stromal component affected by p16. The second option is to present the untreated lung data after the endothelial portion, in a greatly condensed and streamlined manner, as long as a few definitive conclusions (especially about immune cells) can be discerned from this part of the work.

Author Response

We thank the reviewers for their time and effort and believe that addressing their comments has improved the manuscript. We hope that with these additions the manuscript is acceptable for publication.

"Specifically, Figure 1 carries the main message of the paper that palbo pretreatment stimulates lung metastasis via the induction of senescent cells. However, subsequent analysis of the transcriptomic effects of palbo on tumor-free lungs (Figures 2-3) appears to contradict the picture that emerges from the data in Fig. 1.  There seems to be no evidence that palbo induces senescence or SASP in the lungs. Strikingly, palbo apparently does not induce the expression of any genes, the pre-requisite for SASP."

We apologize that this was confusing. We have now added additional gene set enrichment analysis (GSEA) data to the treated normal lung experiments described in Figure 2. Specifically, we have added new GSEA analysis results from Reactome database pathways that are associated with cellular senescence and updated the text accordingly. Intriguingly, in the treated normal ATTAC mouse lungs, four senescence-related gene sets were found to have been significantly positively enriched but only after palbociclib treatment and a drug holiday (revised Figure 2C). This late-developing positive enrichment in senescence-related gene sets in the lungs is then ameliorated by adding AP.

"In fact, it appears that AP senolysis, which should have attenuated the paracrine effects of senescence, by itself induces gene expression, including chemoikines and cytokines, apparently the opposite to what’s expected."

We have added a new Supplemental Figure 3, which is a DAVID gene ontology enrichment analysis of the 68 genes that were significantly upregulated by 10 days of palbociclib + AP in treated normal ATTAC mouse lungs. The most significantly enriched gene ontology terms in these upregulated genes revolve around neutrophil infiltration. This suggests that the upregulated genes likely reflect an innate immune response being mounted against the apotosing senescent cells that happens in parallel to the palbociclib induced changes in macrophage/monocyte levels. We commented on this in the text and updated the remaining supplemental figure numbering accordingly.

"A tenuous link to the overall concept comes from TGF-b pathway observations, but it is not clear if TGF-b effects involve anything that can be biologically linked to the promotion of metastases."

TGFβ pathway activation, which develops in the lungs only after palbociclib treatment plus a drug holiday (Figure 2D), is the only gene set whose enrichment correlates with the late-developing senescence gene set enrichment that we have added as new figure 2C. The late-developing TGFβ pathway enrichment and senescence-related pathway enrichment further correlate as they are all negated by the addition of AP.

"However, the story, as presented, is rather confusing and even appears at times to be self-contradictory, which raises unanswered questions and detracts from the value of this very important work."

We have added text to the conclusion section for the submission to help clarify our proposed model.

Round 2

Reviewer 1 Report

The revised manuscript is acceptable to me. The authors responded to reviewers concerns and suggestions seriously and the revised manuscript is much improved.

Reviewer 2 Report

The additional analysis in the revised MS has closed the gap between the biological observations and the RNA-Seq data in the lungs, making the whole story consistent and revolving around palbo-induced senescence. I recommend acceptance of this detailed and interesting paper.